# Discovering Hidden Algebraic Structures via Transformers with Rank-Aware Beam GRPO

## Abstract

Recent efforts have extended the capabilities of transformers in logical reasoning and symbolic computations. In this work, we investigate their capacity for non-linear latent pattern discovery in the context of functional decomposition, focusing on the challenging algebraic task of multivariate polynomial decomposition. This problem, with widespread applications in science and engineering, is proved to be NP-hard, and demands both precision and insight. Our contributions are threefold: First, we develop a synthetic data generation pipeline providing fine-grained control over problem complexity. Second, we train transformer models via supervised learning and evaluate them across four key dimensions involving scaling behavior and generalizability. Third, we propose Beam Grouped Relative Policy Optimization (BGRPO), a rank-aware reinforcement learning method suitable for hard algebraic problems. Finetuning with BGRPO improves accuracy while reducing beam width by up to half, resulting in approximately 75% lower inference compute. Additionally, our model demonstrates competitive performance in polynomial simplification, outperforming Mathematica in various cases.

## 1 Introduction

Transformers, initially developed for natural language processing Vaswani et al. (2017), have shown remarkable versatility across diverse domains such as vision Dosovitskiy et al. (2020) and protein folding Jumper et al. (2021). More recently, their applications in formal reasoning, symbolic mathematics and algorithmic tasks start to gain traction. Several works have showcased transformer-based architectures' ability to tackle highly structured problems, including theorem proving Polu & Sutskever (2020); Trinh et al. (2024), integration Lample & Charton (2020), matrix multiplication Fawzi et al. (2022) and equation solving Drori et al. (2022).

In this work, we investigate the transformer's capacity for non-linear latent pattern discovery in the context of functional decomposition, i.e. decomposing a complex function as the composition of simpler sub-functions. In contrast to step-by-step logical deduction, or pattern recognition in data analysis, functional decomposition poses significant new challenges to the transformer, because the forms of the sub-functions that we try to discover can be totally hidden or obscured in the final compact form of the original function. Furthermore, it requires extreme precision without any margin of error. Unlike more forgiving classification tasks, the decomposition problem admits only a sparse set of correct solutions: even minor deviations in signs or coefficients can render outputs completely invalid.

Beyond its theoretical interest, functional decomposition has ubiquitous applications in software engineering Tempero et al. (2024), systems biology Mori et al. (2023), mechanical design She et al. (2024), systems engineering Hernandez et al. (2024) and digital logic design Adamski et al. (2005); Lin et al. (2008), where capturing hidden substructures within high-dimensional functions leads to more tractable and efficient models. However, identifying a function's latent compositional structure requires models to look past surface-level correlations, attending instead to deep algebraic symmetries and invariants.

A particularly rich case of functional decomposition arises in multivariate polynomial functions. The polynomial decomposition problem over a ring $k$ seeks to decompose a given polynomial $f \in k[x_1, \ldots, x_n]$ into polynomials $g \in k[y_1, \ldots, y_m]$ and $h_1, \ldots, h_m \in k[x_1, \ldots, x_n]$ such that

$$f(x_1, \ldots, x_n) = g\big(h_1(x_1, \ldots, x_n), \ldots, h_m(x_1, \ldots, x_n)\big). \tag{1}$$

It has wide-ranging applications from cryptography Patarin & Goubin (1997) to dynamical modeling Dang & Testylier (2012), signal processing Demirtas et al. (2012) and robotics Elias & Wen (2025); Manocha & Canny (1992).

The multivariate polynomial decomposition problem has been proved to be NP-hard by Dickerson Dickerson (1987; 1993), although efficient algorithms for various special cases are discussed in Gathen et al. (2003); Von Zur Gathen (1990a;b); Faugère & Perret (2009a;b); Zhao et al. (2012). Crucially, all existing methods either require multiple polynomials as input (system-based approaches) or are restricted to univariate/special structural cases. We are the first to address single-polynomial multi-multivariate decomposition over integers, a problem class for which no general algorithms exist.

To illustrate the difficulty of the problem for the models, let us consider the following expression

$$f = 2a_1^3b_1^3 + 25a_1^2b_1^2 + 6a_1^2a_2b_2b_1^2 + 6a_1^2a_3b_3b_1^2 + 6a_1a_2^2b_2^2b_1 + 6a_1a_3^2b_3^2b_1$$
$$+ 96a_1b_1 + 50a_1a_2b_2b_1 + 50a_1a_3b_3b_1 + 12a_1a_2a_3b_2b_3b_1 + 2a_2^3b_2^3 + 2a_3^3b_3^3$$
$$+ 25a_2^2b_2^2 + 25a_3^2b_3^2 + 6a_2a_3^2b_2b_3^2 + 96a_2b_2 + 6a_2^2a_3b_2^2b_3 + 96a_3b_3$$
$$+ 50a_2a_3b_2b_3 + 128$$

It has a hidden $O(3)$-symmetry, which can be revealed by decomposing $f = g \circ h$, with $g(y) = y^2 + 2(4 + y)^3$ and $h = a_1b_1 + a_2b_2 + a_3b_3$. This is a highly nontrivial task to identify the inner function $h$ directly from the expanded form of $f$, as its structure becomes completely obscured after polynomial substitution, expansion and simplification. Even in this relatively constrained case where $g$ is univariate, discovering the decomposition requires recognizing non-linear latent patterns across dozens of terms. When $g$ becomes multivariate, the complexity increases substantially, making the problem even more challenging.

To tackle the polynomial decomposition problem, we develop a systematic approach with four key components. First, we create a backward synthetic data generation pipeline that allows fine-grained control over polynomial complexity involving range of coefficients, degree, and number of variables. Second, we train lightweight transformer models on these synthetic datasets using supervised learning and analyze how performance scales across four axes (performance complexity scaling, architecture scaling, distribution adaptation, search strategy analysis). Third, we discover that both multi-sampling and greedy search methods struggle with the sparse solution space of the polynomial decomposition problem, and we implement a beam search strategy to effectively extract the models' capabilities. Finally, to address the computational intensity of beam search, we develop a rank-aware variant of the Grouped Relative Policy Optimization (GRPO) reinforcement learning algorithm, which encodes rank information directly in the reward function.

While our systematic evaluation employs synthetic data for controlled complexity analysis, the approach extends to real-world domain-specific decompositions. Applications in crystal field theory (extracting symmetry coefficients), robotics (polynomial barrier certificates), and error correction (syndrome polynomial factorization) typically involve additional structural constraints that make them more tractable than our general case. Our supplementary experiments on $O(N)$ singlet identification achieved 100% accuracy, demonstrating successful transfer to structured problems representative of these applications.

Our study makes the following contributions to neural approaches for polynomial decomposition. First, our backward data generation pipeline enables targeted training across varying levels of decomposition difficulty. Second, our comprehensive evaluation across four dimensions, for the first time, establishes robust baselines for transformers' performance on polynomial decomposition tasks. Third, using the rank-aware Beam Grouped Relative Policy Optimization (BGRPO), our models improve accuracy while reducing beam search width by up to 50%, resulting in 75% lower computational requirements during inference. Additionally, our model demonstrates competitive performance in polynomial simplification, outperforming Mathematica in various cases. This underscores the potential of neural models to complement and extend classical symbolic computation capabilities.

## 2 METHOD

### 2.1 BACKWARD SYNTHETIC DATA GENERATION

We generate synthetic data for supervised learning using a backward approach, starting from the decomposed form. First, we generate the inner functions ($h_1, \ldots, h_m$ in Eq. equation 1) and the outer function ($g$ in Eq. equation 1) with random monomial terms of bounded degree and random coefficients within a given range. Then, we obtain the composed function ($f$ in Eq. equation 1) via substitution, expansion, and term collection. See Appendix A for the detailed algorithm. For each generated instance, we create a training pair consisting of the expanded polynomial $f$ as input and its decomposed components $\{g, h_1, \ldots, h_{v_{\text{outer}}}\}$ as the target output. The model is trained to minimize the standard negative log-likelihood loss function.

Our synthetic data generation process provides fine-grained control over problem complexity through eight parameters: $C_{\text{inner}}$ (coefficient range for inner polynomials), $d_{\text{inner}}$ (maximum degree of inner polynomials), $v_{\text{inner}}$ (number of variables in inner polynomials), $t_{\text{inner}}$ (maximum number of terms in inner polynomials), and similarly $C_{\text{outer}}$, $d_{\text{outer}}$, $v_{\text{outer}}$, and $t_{\text{outer}}$ for the outer polynomial.

## 2.2 BEAM SEARCH

Beam search is a breadth-first search algorithm that approximates optimal decoding by keeping track of the $k$ most probable sequences at each step Freitag & Al-Onaizan (2017). For each of the $k$ current sequences, the algorithm considers the top-$k$ token extensions per sequence. These $k^2$ candidate continuations are then ranked by the sum of log probabilities of all tokens in the sequence, and only the top-$k$ sequences with the highest cumulative log probability are retained for the next step. In this paper, we refer to $k$ as the beam width, and to the position (1st, 2nd, etc.) of an output in the final beam as its rank.

Our analysis across all model outputs identified a specific error pattern in polynomial decomposition: the model achieves approximately 90% accuracy for predicting non-sign tokens (operators, numbers, variables), but exhibits near-random performance for deciding between positive and negative signs. This creates a unique inference challenge where exploration needs to be constrained for high-confidence structural elements while simultaneously expanded for uncertain sign choices.

Beam search is particularly well-suited for this situation as it maintains the high-confidence structural backbone while systematically exploring variations in the uncertain components. Our experiments demonstrate that beam search significantly outperforms greedy decoding and random sampling for polynomial decomposition tasks. See Appendix C for a detailed error analysis and an explanation of beam search effectiveness for this task.

## 2.3 BGRPO : REINFORCEMENT LEARNING METHOD ENHANCING BEAM SEARCH EFFICIENCY

The computational cost of beam search scales quadratically with beam width. There would be a significant computational advantage if we could improve the ranks of correct outputs. To address this, we introduce Beam Grouped Relative Policy Optimization (BGRPO), a reinforcement learning method that extends GRPO, uniquely taking into account rankings in the beam search, specifically designed for improving beam search inference efficiency.

Traditional RL methods like PPO Schulman et al. (2017) and standard GRPO create a training-inference mismatch: they train on randomly sampled outputs but deploy beam search at inference. BGRPO addresses this by incorporating beam search directly into the training loop, aligning training with deployment. While GRPO assumes independent samples for baseline calculation, beam search generates correlated outputs that share high-confidence structural elements but differ in uncertain components.

Reinforcement learning enables models to explore solution spaces more effectively than supervised learning alone, enhancing the model's capabilities by addressing specific weaknesses through a reward mechanism. This approach encourages correct answers while discouraging incorrect ones based on an advantage function—the difference between a solution's reward and a baseline reward. Group Relative Policy Optimization (GRPO) Shao et al. (2024) estimates this baseline for each question by sampling a group of outputs, and has shown promising results for reinforcement learning in language generation tasks due to its sample efficiency and stability DeepSeek-AI (2025).

Our proposed Beam Grouped Relative Policy Optimization (BGRPO) extends this approach by using beam search rather than independent sampling for generating the group of outputs. While this significantly alters the distribution of outputs, making their average reward less suitable as a traditional baseline, it still provides valid training signals by reinforcing correct answers and penalizing incorrect ones. BGRPO is particularly effective for our task because beam search generates outputs with identical structure that differ only in the confusing elements (signs), creating a focused learning signal.

Additionally, BGRPO incorporates rank information directly into the reward function by applying an exponential decay factor based on the position in the beam. This incentivizes correct answers to appear at earlier positions in the beam search, effectively pushing correct solutions toward the top of the beam ranking.

**Training Objective** For a prompt $x$, let $\mathcal{B}(x) = \{y_1, \ldots, y_w\}$ be the set of beam search outputs with beam width $w$ generated by the old policy $\pi_{\theta_{\text{old}}}$. Each output sequence $y_i$ receives a reward $r_i$, where $r_i = 0$ for incorrect polynomial decomposition and $r_i = 1$ for correct decomposition. In BGRPO, we incorporate rank information by scaling the

reward for correct decompositions using an exponential decay function $e^{-\text{rank}/w}$. We optimize the policy model $\pi_\theta$ for $\mu$ iterations by maximizing the following objective:

$$\mathcal{J}_{\text{BGRPO}}(\theta) = \frac{1}{w} \sum_{i=1}^{w} \left( \min \left( \frac{\pi_\theta(y_i|x)}{\pi_{\theta_{\text{old}}}(y_i|x)} A_i, \text{clip} \left( \frac{\pi_\theta(y_i|x)}{\pi_{\theta_{\text{old}}}(y_i|x)}, 1 - \varepsilon, 1 + \varepsilon \right) A_i \right) - \beta \mathbb{D}_{\text{KL}}(\pi_\theta || \pi_{\text{ref}}) \right), \quad (2)$$

where $\varepsilon$ is the clipping parameter that constrains policy updates and $\beta$ controls the KL divergence regularization term:

$$\mathbb{D}_{\text{KL}}(\pi_\theta || \pi_{\text{ref}}) = \frac{\pi_{\text{ref}}(o_i|q)}{\pi_\theta(o_i|q)} - \log \frac{\pi_{\text{ref}}(o_i|q)}{\pi_\theta(o_i|q)} - 1. \quad (3)$$

Here, $\pi_{\text{ref}}$ is the reference policy, which is the initial model before BGRPO training. The advantage function $A_i$ is computed without normalization as $A_i = r_i - \text{mean}(\{r_1, r_2, \cdots, r_w\})$, following the approach in Liu et al. (2025).

## 3 EXPERIMENTAL SETUP

### 3.1 EVALUATION AXES

To systematically analyze our models' capabilities for the polynomial decomposition problem, we consider four key evaluation dimensions.

**Problem Complexity Scaling ($\mathcal{D}_1$).** We analyze how the model performance varies with respect to changes in the complexity parameters for synthetic data generation. We vary the number of variables $v_{\text{inner}}$, $v_{\text{outer}}$, and the maximum degrees $d_{\text{inner}}, d_{\text{outer}}$ for both the inner and outer polynomials.

**Architecture Scaling ($\mathcal{D}_2$).** We investigate how model performance scales with key architectural hyperparameters of the transformer. In particular, we measure $\mathcal{P}(M(d, l, a))$, the performance of models with embedding dimension $d$, number of layers $l$, and number of attention heads $a$. Our goal is to characterize how these hyperparameters influence model capabilities.

**Distribution Adaptation ($\mathcal{D}_3$).** A practical challenge in applying transformers to symbolic computation is their sensitivity to the numerical ranges present in the training data. For example, models trained on specific coefficient ranges tend to struggle with polynomials outside these ranges. On the other hand, we found that models can rapidly adapt to new coefficient distributions with minimal additional training, suggesting that they manage to learn generalizable pattern recognition rather than merely memorizing specific numerical relationships.

To quantify the model's ability to transfer its polynomial decomposition skills to numerically distinct but structurally identical problems, we prepare the model $M_{C_1 \to C_2}^n$. This model is initially trained on 1M polynomial decomposition examples with $C_{\text{outer}} = C_1$ and then fine-tuned with $n$ examples with $C_{\text{outer}} = C_2$ where $C_1 \cap C_2 = \emptyset$. We measure the performance of model $M_{C_1 \to C_2}^n$ on a test set of polynomial decomposition problems with $C_{\text{outer}} = C_2$:

$$\mathcal{G}(n) = \mathcal{P}\left(M_{C_1 \to C_2}^n, \text{ test set with } C_{\text{outer}} = C_2\right) \quad (4)$$

**Search Strategy Analysis ($\mathcal{D}_4$).** We investigate how beam search enhances model performance on polynomial decomposition tasks, analyzing its effectiveness across different model architectures and levels of problem complexity.

### 3.2 SYNTHETIC DATASET SETUP

For the axis $\mathcal{D}_1$ of the problem complexity scaling, we first examine degree scaling by training a model on 2M polynomial decomposition examples with different inner and outer degrees as described in Table 1. We then evaluate this model on separate test datasets with the same configuration parameters, each corresponding to one of nine different $(d_{\text{inner}}, d_{\text{outer}})$ pairs to assess performance across varying problem complexities.

For the second part of the $\mathcal{D}_1$ axis, we train a model for each combination of $v_{\text{inner}}$ and $v_{\text{outer}}$ varying from 2 to 4 while fixing the other parameter at 3. For each combination, we use 1M examples to train the model.

For the axis $\mathcal{D}_2$ of architecture scaling, we train multiple models with varying architectural configurations, all using the same dataset of 2M examples with polynomial parameters as described in Table 1.

For the axis $\mathcal{D}_3$ of distribution adaptation, we train initial models on 1M examples with $C_{\text{outer}} = C_1 = [-5, 5]$ and then adapt them to examples with $C_{\text{outer}} = C_2 = [-10, -6] \cup [6, 10]$. Other parameters are the same across both datasets as described in Table 1.

For the second part of $\mathcal{D}_1$ (Variable Scaling) and $\mathcal{D}_2$, we set $t_{\text{inner}} = t_{\text{outer}} = 3$ to prevent expressions from becoming too long. We describe our tokenization in Appendix B.

Table 1: Synthetic Dataset Configuration Across Evaluation Axes

| Evaluation Axis | Inner Coeff. | Outer Coeff. | Inner Degrees | Outer Degrees | Inner Vars | Outer Vars |
|---|---|---|---|---|---|---|
| $\mathcal{D}_1$ (Degree Scaling) | $[-20, 20]$ | $[-20, 20]$ | $\{2, 3, 4\}$ | $\{2, 3, 4\}$ | 1 | 1 |
| $\mathcal{D}_1$ (Variable Scaling) | $[-5, 5]$ | $[-5, 5]$ | 3 | 3 | $\{2, 3, 4\}$ | $\{2, 3, 4\}$ |
| $\mathcal{D}_2$ (Architecture) | $[-5, 5]$ | $[-5, 5]$ | 3 | 3 | 3 | 3 |
| $\mathcal{D}_3$ (Adaptation) | $[-20, 20]$ | $C_1 = [-5, 5]$ | $\{1, 2\}$ | $\{1, 2, 3, 4\}$ | 1 | 1 |
| | $[-20, 20]$ | $C_2 = [-10, -6] \cup [6, 10]$ | $\{1, 2\}$ | $\{1, 2, 3, 4\}$ | 1 | 1 |

## 3.3 ARCHITECTURE CONFIGURATION

We employ a decoder-only transformer architecture following standard design principles Vaswani et al. (2017). Table 2 summarizes our task-specific configurations across all experimental axes. For lightweight and effective training, we developed our own model and training pipeline based on `minGPT` Karpathy (2020).

Table 2: Transformer Model Configuration Across Experiments

| Experiment | Context Window | Embedding Dim. | Layers | Heads |
|---|---|---|---|---|
| $\mathcal{D}_1$ (Degree Scaling) | 256 | 512 | 6 | 8 |
| $\mathcal{D}_1$ (Variable Scaling) | 850 | 512 | 6 | 8 |
| $\mathcal{D}_2$ (Architecture) | 850 | $\{256, 512, 768\}$ | $\{4, 6\}$ | 8 |
| $\mathcal{D}_2$ (Attention Heads) | 850 | 512 | 6 | $\{4, 8, 16\}$ |
| $\mathcal{D}_3$ (Distribution) | 256 | 512 | 4 | 8 |

*Common settings: GELU activation, learned positional embeddings, multi-head attention with causal masking, MLP hidden dimension = $4\times$ embedding dimension.*

## 3.4 SUPERVISED LEARNING DETAILS

We train our models using the Adam optimizer with an initial learning rate of $6 \times 10^{-4}$, incorporating a 10% warmup period followed by cosine decay. Each configuration initially trains on 1M instances, with additional 1M training examples added incrementally until performance saturation. We use a batch size of 200 throughout training. We train models with enough epochs until it saturates with the given dataset.

## 3.5 BGRPO IMPLEMENTATION

For the BGRPO reinforcement learning phase, we generate candidate solutions using beam search with a width of 32 and temperature of 1.0. We implement our approach using the GRPO functionality from the `trl` library von Werra et al. (2020). The training process consists of 5 policy update iterations after sampling outputs for 8 distinct polynomial decomposition problems. We set the PPO clipping parameter $\varepsilon$ to 0.2 and the KL divergence coefficient $\beta$ to 0.01. The learning rate during BGRPO training is $1 \times 10^{-5}$. We train models from $\mathcal{D}_2$ on a dataset of 200 non-repeating problems, saving checkpoints every 5 iterations and selecting the best model based on performance with beam width 7.

# 4 EXPERIMENTAL RESULTS

## 4.1 PROBLEM COMPLEXITY SCALING ($\mathcal{D}_1$)

In the first part of $\mathcal{D}_1$, we examine how model performance varies with the degrees of inner and outer polynomials. The result is shown in Figure 1. We use greedy search for the inference. Regardless of the degrees of the polynomials, our model achieves a remarkable single-output accuracy. Notably, when using beam search with a width of 10, the model's accuracy reaches 100% for these configurations.

Our analysis reveals a pattern: performance remains invariant to increases in the outer polynomial's degree, while decreasing when the inner polynomial's degree increases. This demonstrates that the transformer's decomposition capability is primarily limited by the complexity of the inner polynomial rather than that of the outer polynomial.

In the second part of $\mathcal{D}_1$, we investigate how the performance scales with $v_{\mathrm{inner}}$ and $v_{\mathrm{outer}}$, the number of variables in the inner and outer polynomials. Figures 2 and 3 present these results.

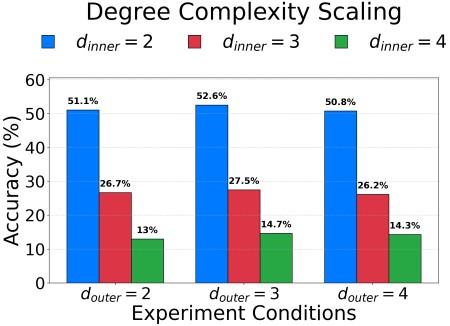

Figure 1: Performance across different $d_{\mathrm{inner}}, d_{\mathrm{outer}}$

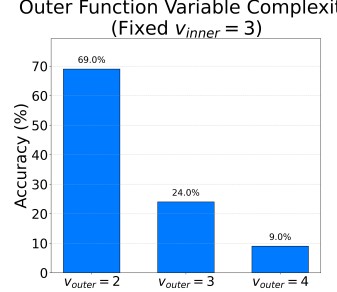

Figure 2: Performance across different $v_{\mathrm{outer}}$

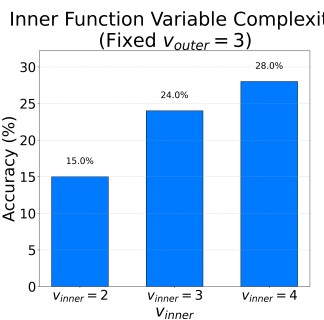

Figure 3: Performance across different $v_{\mathrm{inner}}$

Given the challenging nature of multivariate polynomial decomposition, we evaluate the model's performance using beam search with a width of 30, considering a prediction correct if at least one of the 30 candidate outputs is correct decomposition.

Our results reveal two trends: performance decreases dramatically as $v_{\mathrm{outer}}$ increases, yet counter-intuitively improves as $v_{\mathrm{inner}}$ increases. This observation aligns with the following heuristic understanding: higher $v_{\mathrm{outer}}$ creates an information bottleneck, requiring the model to simultaneously resolve multiple interdependent inner functions. In contrast, higher $v_{\mathrm{inner}}$ provides more dimensions of input variation with additional structural indicators that can guide the decomposition process.

## 4.2 ARCHITECTURE SCALING ($\mathcal{D}_2$)

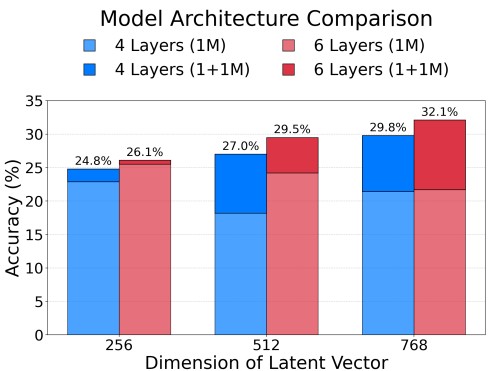

Figure 4: Accuracies on different number of layer and dimension.

In $\mathcal{D}_2$, we examine how model performance varies with architectural parameters: embedding dimension, number of layers, and number of attention heads. When varying the number of heads, we maintain a constant total embedding dimension, meaning that models with more heads have smaller per-head embedding dimensions. We use the dataset described in Section 3.2 and evaluate using beam search with a width of 30.

Figure 4 reveals the scaling behavior Kaplan et al. (2020) of transformer architectures on polynomial decomposition. As model capacity increases through higher embedding dimensions and additional layers, performance consistently improves.

Notably, our results demonstrate the presence of a data-dependent scaling threshold. With limited training data (1M examples), larger models

initially underperform their simpler counterparts, particularly evident in the 6-layer configurations with higher embedding dimensions. However, this pattern reverses completely with additional training data, confirming that larger models possess superior capacity for mathematical pattern recognition when provided with sufficient examples to leverage their parametric advantage.

In $\mathcal{D}_2$, we also examine model performance with different numbers of attention heads. Our experiments reveal that increasing the number of attention heads while maintaining constant total embedding dimension leads to progressively deteriorating performance on polynomial decomposition tasks. Models with 4 heads achieved 32.0% accuracy, while those with 8 and 16 heads reached only 28.0% and 25.0% accuracy, respectively. This suggests that for our specific task of mathematical pattern recognition, fewer, more expressive attention heads with larger per-head dimensions provide better performance than numerous specialized heads with smaller dimensions.

### 4.3 DISTRIBUTION ADAPTATION ($\mathcal{D}_3$)

We evaluate $\mathcal{G}(n)$ as defined in Eq. 4, which measures how quickly models adapt to new coefficient distributions as a function of adaptation sample size $n$. For this experiment, we train a model with 4 layers and 512 embedding dimension on the dataset described in Section 3.2. The initial training used 1M examples with outer polynomial coefficient range $C_1$, followed by fine-tuning on $n$ examples with coefficient range $C_2$ for a single epoch. We report the variance in accuracy based on three independent trials.

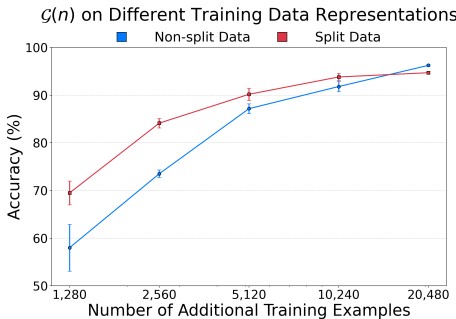

Figure 5: Performance recovery when adapting to a new coefficient distribution

Models trained exclusively on the first dataset achieve only 5.67% accuracy on the new distribution, despite reaching nearly 100% accuracy on the original distribution. Figure 5 illustrates how performance recovers during adaptation. Notably, despite using only $\approx 2\%$ of the original training data size, the model rapidly recovers its accuracy from single digits to over 90%. This rapid adaptation indicates successful transfer learning, suggesting that the model develops a general mathematical understanding of polynomial substructures rather than memorizing specific numerical relationships.

We further investigate whether alternative data representations could enhance this adaptation capability. We propose "split" representation of polynomials, where we randomly select terms from the expanded form and split their coefficients. For example:

$$f_{\text{non-split}}(a) = -63 + 23a - 71a^2 - 11a^3 - 14a^4 - 12a^5 - 2a^6$$
$$f_{\text{split}}(a) = -63 + 23a - 4a^2 - 67a^2 - 8a^3 - 3a^3 - 7a^4 - 7a^4 - 12a^5 - a^6 - a^6 \tag{5}$$

In Figure 5, the red line demonstrates $\mathcal{G}(n)$ of the model trained on data with both normal and split representation. Models trained on this mixed data including split representation demonstrate significantly faster adaptation, requiring only 70% of the additional training examples to reach equivalent performance on the new distribution.

This enhanced generalization likely stems from the model being forced to recognize mathematically equivalent but differently represented polynomials, compelling it to develop a deeper understanding of polynomial structure rather than memorizing specific patterns.

### 4.4 SEARCH STRATEGY ANALYSIS ($\mathcal{D}_4$)

We evaluate how search strategies impact model performance on polynomial decomposition tasks, with a particular focus on beam search efficiency. Figure 6 and 7 illustrate the accuracy achieved across different beam widths for polynomials with varying numbers of variables.

Our results reveal an unusually dramatic impact of beam search for polynomial decomposition compared to typical NLP tasks. For two-variable polynomials, accuracy improves from 11% with greedy search to 69% with a beam width of

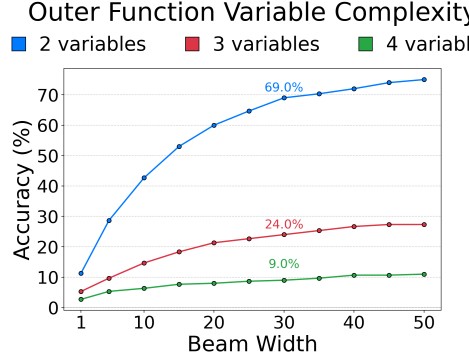

Figure 6: Beam width scaling with varying $v_{\text{outer}}$ ($v_{\text{inner}} = 3$)

Figure 7: Beam width scaling with varying $v_{\text{inner}}$ ($v_{\text{outer}} = 3$)

30—a remarkable 6.3× improvement. This stands in stark contrast to standard neural machine translation applications, where beam search typically yields BLEU score improvements of only 2-4 points Huang et al. (2018); Ranzato et al. (2016). Even more telling, most NMT systems show diminishing returns with beam widths beyond 5-10 Freitag & Al-Onaizan (2017).

### 4.5 BGRPO Results

We evaluated BGRPO across models of varying sizes from our architecture scaling experiments($\mathcal{D}_2$), implementing versions both with and without rank signal. Fig 8 illustrates these results.

BGRPO consistently improved accuracy across all beam widths regardless of model size. Without rank signal, BGRPO gives average accuracy increases of 34.0%, 17.8%, and 12.4% for 6-layer models with dimension 256, 512, and 768 respectively. Including rank signal in BGRPO produces even more improvements, with average accuracy increases of 46.6%, 28.4%, and 30.2%.

These improvements translate to significant computational efficiency gains. For instance, the dimension-256 model initially achieved 26.1% accuracy with beam width 30. After applying BGRPO with rank signal, comparable accuracy (26.0%) was achieved with just beam width 16. This effectively halves the required beam width for equivalent performance. Since beam search computation scales quadratically with beam width, this improvement reduces beam search computation by approximately 75% while maintaining equivalent performance.

On average, BGRPO without rank signal reduced the required beam width by 31.3%, 14.9%, and 11.4% for 6-layer models with dimension 256, 512, and 768 respectively. When incorporating rank signal, BGRPO reduced required beam width even further, by 38.9%, 22.0%, and 26.5%.

### 4.6 Simplification Comparison with Mathematica

While polynomial simplification and polynomial decomposition represent two distinct mathematical objectives, simplification frequently arises as a consequence of decomposition, since decomposed forms generally exhibit reduced algebraic complexity compared to the original expression. In this subsection, we briefly explore the capabilities of our models for this related problem, and benchmark against the most powerful symbolic computation engine Mathematica. Despite our lightweight parameter budgets and the absence of any explicit simplification objective in our training, the models were able to reduce the leaf count Wolfram Research, Inc. (1996) of complex expressions, with performance on par with — and in two of five complexity regimes surpassing —Mathematica's state-of-the-art FullSimplify function (see Table 3, competitive performances are bolded).

These findings highlight that transformers' inherent ability to uncover latent patterns rivals that of the most advanced symbolic computation methods.

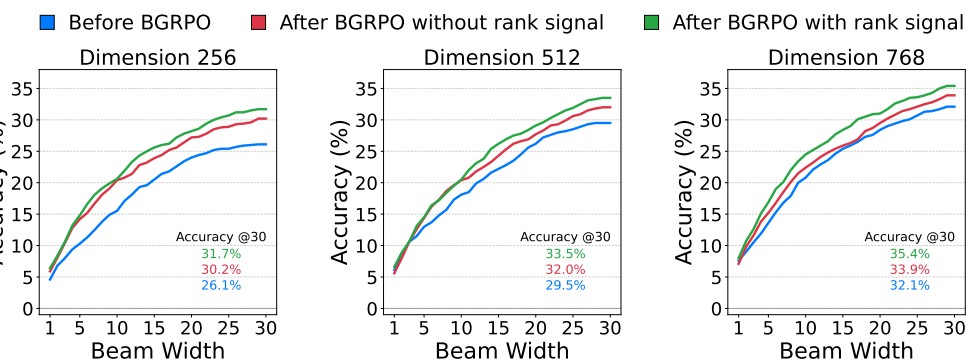

Figure 8: Accuracies on experiments with different dimension. Each experiment we have finetuned model with 2M data and models trained with BGRPO with and without rank signal on top of that.

Table 3: Average leaf count comparison (Beam width $= 30$)

| Problem Complexity | | Leaf Count (mean) | | |
|---|---|---|---|---|
| $v_O$ | $v_S$ | Transformer | Mathematica | $\Delta$ |
| 2 | 3 | **27.28** | 30.03 | **-2.75** |
| 3 | 3 | 22.85 | **22.12** | **0.73** |
| 4 | 3 | 22.52 | 20.00 | 2.52 |
| 3 | 2 | 17.27 | **17.10** | **0.17** |
| 3 | 4 | **26.04** | 27.56 | **-1.52** |

## 5  CONCLUSION

Our investigation into transformers for polynomial decomposition uncovers key insights into how neural networks can infer hidden algebraic structures.

We find that model performance depends asymmetrically on polynomial complexity parameters ($\mathcal{D}_1$): inner polynomial degree plays a dominant role, while outer polynomial complexity has limited impact. Counterintuitively, increasing the number of inner variables improves accuracy by imposing structural constraints, whereas more outer variables create information bottlenecks.

From an architectural viewpoint ($\mathcal{D}_2$), we confirm that performance scales with model size. We observe that fewer but more expressive attention heads are especially effective for this task. In terms of distribution adaptation ($\mathcal{D}_3$), models transfer rapidly to new coefficient distributions, requiring as little as 2% of the original training data, indicating that they internalize generalizable principles rather than rely on memorization. Moreover, we can enhance this generalization capability through strategic dataset design.

Beam search analysis ($\mathcal{D}_4$) yields up to 6.3× improvement over greedy decoding due to the sparse, precise nature of mathematical solutions. Models finetuned with our rank-aware BGRPO reinforcement learning method achieve equivalent accuracy with up to 50% smaller beam widths, cutting inference computation by approximately 75%. Lastly, our model demonstrates competitive performance in polynomial simplification compared with symbolic computation tools in Mathematica.

Our work provides, for the first time, a systematic analysis of transformer capabilities for polynomial decomposition through carefully controlled experiments across four dimensions. Our methodologies can serve as a road map for exploring neural models in other domains that require non-local latent pattern discovery, such as functional decomposition problems ranging from systems engineering and mechanical design to digital logic design. While we developed BGRPO specifically for enhancing beam search in the polynomial decomposition problem, similar techniques may prove useful in other domains with sparse solution spaces where models can identify correct structures but struggle with specific details.

ACKNOWLEDGMENTS

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

## A  BACKWARD SYNTHETIC DATA GENERATION ALGORITHM

Our backward synthetic data generation in subsection 2.1 can be described as follows. [H] [1] Coefficient range $C_{\text{inner}}$, $C_{\text{outer}}$; maximal degrees $d_{\text{inner}}$, $d_{\text{outer}}$; variable counts $v_{\text{inner}}$, $v_{\text{outer}}$; term limits $t_{\text{inner}}$, $t_{\text{outer}}$. Generate outer polynomial $g$ with $v_{\text{outer}}$ variables, coefficients $\in C_{\text{outer}}$, degree $= d_{\text{outer}}$, and no more than $t_{\text{outer}}$ monomial terms. Generate $v_{\text{outer}}$ inner polynomials $h_1, \ldots, h_{v_{\text{outer}}}$, where each $h_i$ has $v_{\text{inner}}$ variables, coefficients $\in C_{\text{inner}}$, degree $= d_{\text{inner}}$, and no more than $t_{\text{inner}}$ monomial terms. $f \leftarrow g(h_1, \ldots, h_{v_{\text{outer}}})$, i.e. substitute $h_1, \ldots, h_{v_{\text{outer}}}$ into $g$, expand and collect the monomial terms. **return** $(f,\ g,\ h_1, \ldots, h_{v_{\text{outer}}})$

## B  TOKENIZATION

We encode polynomials using prefix notation, with separate tokens for operators, digits, and variables. Each number includes its sign, so we only use addition, multiplication, and power operators. Subtraction is represented as addition with a negative sign. Each input sequence consists of the tokenized expanded polynomial $f$ followed by a question mark token '?'. The target output format depends on the number of outer variables: for $v_{\text{outer}} = 1$, the target output is simply the tokenized inner polynomial $h$; for $v_{\text{outer}} > 1$, the target output begins with the tokenized outer polynomial $g$ followed by each tokenized inner polynomial $h_1, \ldots, h_{v_{\text{outer}}}$, with all polynomials separated by a delimiter token '&'.

Below is an example of a tokenized training input 'x' and target output 'y':

$x: + * \text{P } 9\, 0\ a + * \text{N } 3\, 1\, 9\ \hat{}\ a\, \text{P } 2 + * \text{N } 3\, 6\ \hat{}\ a\, \text{P } 3 * \text{N } 1\ \hat{}\ a\, \text{P } 4\ ? + \text{N } 5 + * \text{P } 1\, 8\, a\ \hat{}\, a\, \text{P } 2\ \square \ldots$
$y: \square\square\square\square\square\square\square\square\square\square\square\square\square\square\square\square\square\square\square\square\square\square\square\square\square\square\square\square\square\square\square\square\square\square\square\square\square\square\square\square + \text{N } 5 + * \text{P } 1\, 8\, a\ \hat{}\, a\, \text{P } 2\ \square\square \ldots$

This example shows a training pair where the outer polynomial is $90a - 319a^2 - 36a^3 - a^4$ and the target inner polynomial is $-5 + 18a + a^2$. The $\square$ symbol represents a padding token which is excluded from the log-likelihood loss calculation.

## C  EXAMPLE OUPUT LOGITS AND EFFECTIVENESS OF THE BEAM SEARCH

Figure 9 shows example top-3 probabilities for each token position in the answer sequence at temperature 1, using the layer-6, embedding dimension 512 model from our $\mathcal{D}_2$ experiments. Correct answers are highlighted in red. The visualization clearly illustrates that the model's primary source of confusion occurs in sign decisions, while it confidently predicts most of the other token types.

Table 4 quantifies this observation by showing the probability and accuracy statistics for different token types across our model architectures from $\mathcal{D}_2$. These statistics were computed using a test set of 1000 polynomial decomposition problems at temperature 1.

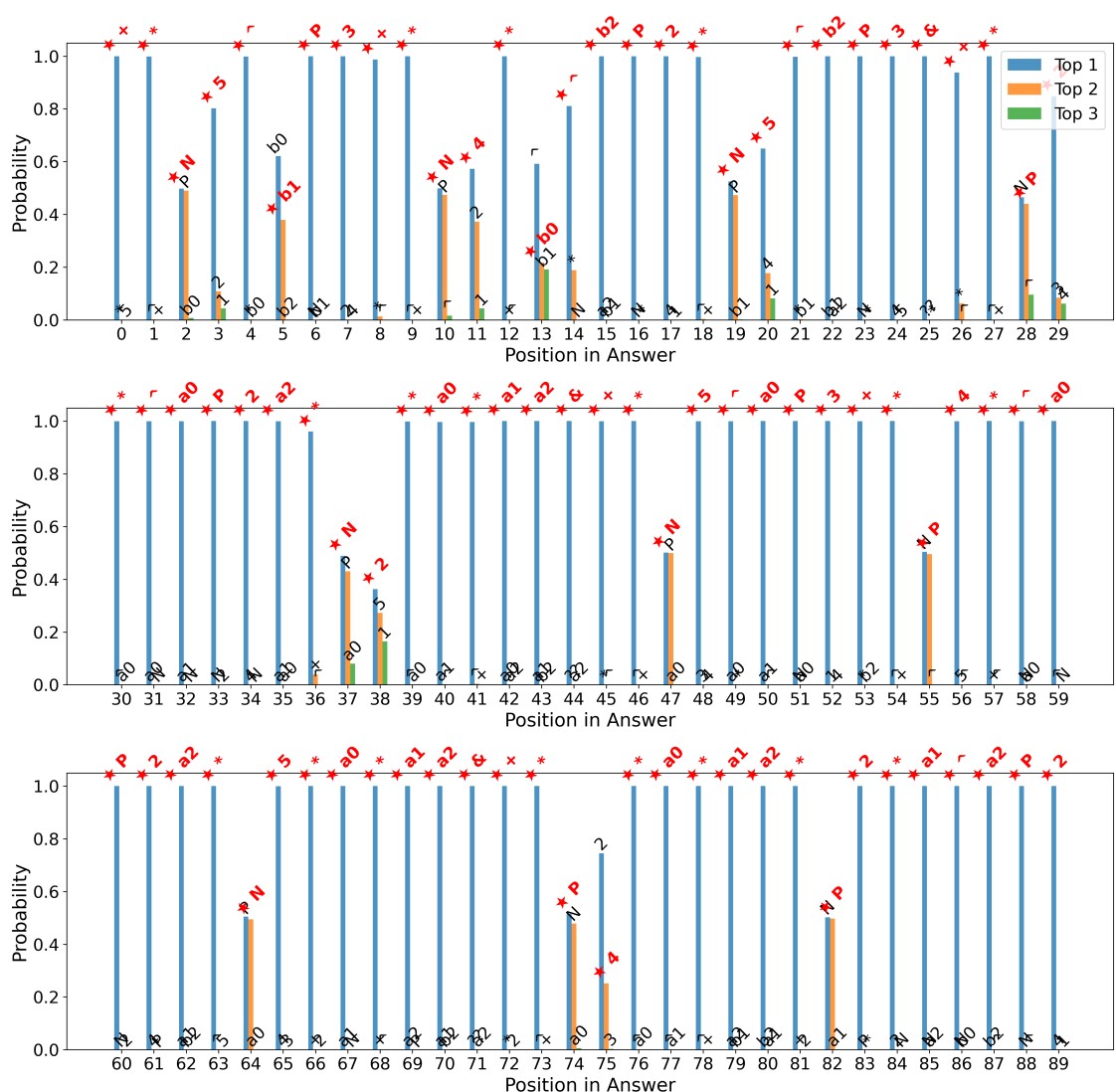

Figure 9: Top-3 probability for each token position in the answer sequence where

**Answer:** + * N 5 ^ b1 P 3 + * N 4 * b0 ^ b2 P 2 * N 5 ^ b2 P 3 & + * P 2 * ^ a0 P 2 a2 * N 2 * a0 * a1 a2 & + * N 5 ^ a0 P 3 + * P 4 * ^ a0 P 2 a2 * N 5 * a0 * a1 a2 & + * P 4 * a0 * a1 a2 * P 2 * a1 ^ a2 P 2

**Question:** + * P 6 2 5 ^ a0 P 9 + * N 1 5 0 0 * ^ a0 P 8 a2 + * P 1 8 7 5 * ^ a0 P 7 * a1 a2 + * P 1 2 0 0 * ^ a0 P 7 ^ a2 P 2 + * N 3 0 0 0 * ^ a0 P 6 * a1 ^ a2 P 2 + * P 1 8 7 5 * ^ a0 P 5 * ^ a1 P 2 ^ a2 P 2 + * N 3 2 0 * ^ a0 P 6 ^ a2 P 3 + * P 1 2 0 0 * ^ a0 P 5 * a1 ^ a2 P 3 + * N 1 6 2 8 * ^ a0 P 4 * ^ a1 P 2 ^ a2 P 3 + * P 4 3 3 * ^ a0 P 3 * ^ a1 P 3 ^ a2 P 3 + * N 1 2 8 * ^ a0 P 3 * ^ a1 P 2 ^ a2 P 4 + * N 3 5 2 * ^ a0 P 2 * ^ a1 P 3 ^ a2 P 4 + * N 3 2 * ^ a0 P 2 * ^ a1 P 2 ^ a2 P 5 + * N 2 0 8 * a0 * ^ a1 P 3 ^ a2 P 5 * N 4 0 * ^ a1 P 3 ^ a2 P 6 ?

Table 4: Token Type Analysis Across Different Model Architectures

| Token Type | Metric | 4 Layers | | | 6 Layers | | |
| --- | --- | --- | --- | --- | --- | --- | --- |
| | | 256 dim | 512 dim | 768 dim | 256 dim | 512 dim | 768 dim |
| Sign | Probability | $0.489 \pm 0.001$ | $0.489 \pm 0.001$ | $0.493 \pm 0.001$ | $0.491 \pm 0.001$ | $0.490 \pm 0.001$ | $0.490 \pm 0.001$ |
| | Accuracy | $0.519 \pm 0.006$ | $0.531 \pm 0.006$ | $0.530 \pm 0.006$ | $0.522 \pm 0.006$ | $0.523 \pm 0.006$ | $0.521 \pm 0.006$ |
| Operator | Probability | $0.920 \pm 0.002$ | $0.915 \pm 0.002$ | $0.919 \pm 0.002$ | $0.927 \pm 0.002$ | $0.925 \pm 0.002$ | $0.925 \pm 0.002$ |
| | Accuracy | $0.937 \pm 0.002$ | $0.934 \pm 0.002$ | $0.935 \pm 0.002$ | $0.943 \pm 0.002$ | $0.941 \pm 0.002$ | $0.942 \pm 0.002$ |
| Number | Probability | $0.880 \pm 0.002$ | $0.870 \pm 0.002$ | $0.878 \pm 0.002$ | $0.890 \pm 0.002$ | $0.885 \pm 0.002$ | $0.884 \pm 0.002$ |
| | Accuracy | $0.901 \pm 0.002$ | $0.893 \pm 0.003$ | $0.897 \pm 0.002$ | $0.911 \pm 0.002$ | $0.905 \pm 0.002$ | $0.903 \pm 0.002$ |

Note: Values shown as mean $\pm$ standard error of the mean. The sign token probabilities are near-random, while operators and numbers show high confidence and accuracy.

As discussed in Section 2.2, our models achieve approximately 90% accuracy when predicting non-sign tokens, but exhibit near-random performance when choosing between positive and negative signs. This specific error pattern makes beam search particularly effective for our task.

The effectiveness of beam search stems from its ability to explore multiple sign configurations while preserving the high-confidence structural tokens. In probability terms, selecting a token with 0.1 probability instead of one with 0.9 probability is equivalent to making approximately 11 consecutive choices of a 0.45 probability token over a 0.55 probability token. Since our polynomial expressions typically contain fewer than 10 sign decisions, beam search with a width of approximately 30 can efficiently cover most viable sign permutations while maintaining the correct monomial structure identified with high confidence.

## D ATTENTION SCORE ANALYSIS: MONOMIAL HEADS

Attention mechanism analysis has provided valuable insights into transformer model behaviors, with studies identifying specialized attention heads that serve specific functions. For example, Olsson et al. (2022) identified "Induction Heads" that play a crucial role in in-context learning, while Wang et al. (2022) provided a comprehensive understanding of indirect object identification in GPT-2 Small.

In our analysis of attention patterns in polynomial decomposition models, we identified specialized attention heads that recognize the structure of polynomials, particularly focusing on monomial identification. We call these "Monomial Heads," and they appear consistently across all model sizes in our architecture scaling experiments ($\mathcal{D}_2$).

Monomial Heads manifest in two distinct patterns in our models. First, in layer 0, several attention heads consistently attend to tokens 1-5 positions behind the current position, as shown in the leftmost plot of Figure 10. Second, in layer 1, we observe specialized behavior where certain heads focus attention on specific tokens within each monomial of the input polynomial (middle plot), while others specifically attend to delimiter tokens in the decomposition output (rightmost plot).

We hypothesize that this represents a two-stage process: in the first layer, the model identifies key tokens that serve as indicators for each monomial by examining local context (1-5 tokens behind). In the second layer, tokens within each monomial attend to these indicator tokens to establish their monomial membership. While this pattern is most clear in the encoding of the input polynomial, the decomposition output shows evidence of boundary recognition, particularly at the transitions between inner functions marked by delimiter tokens.

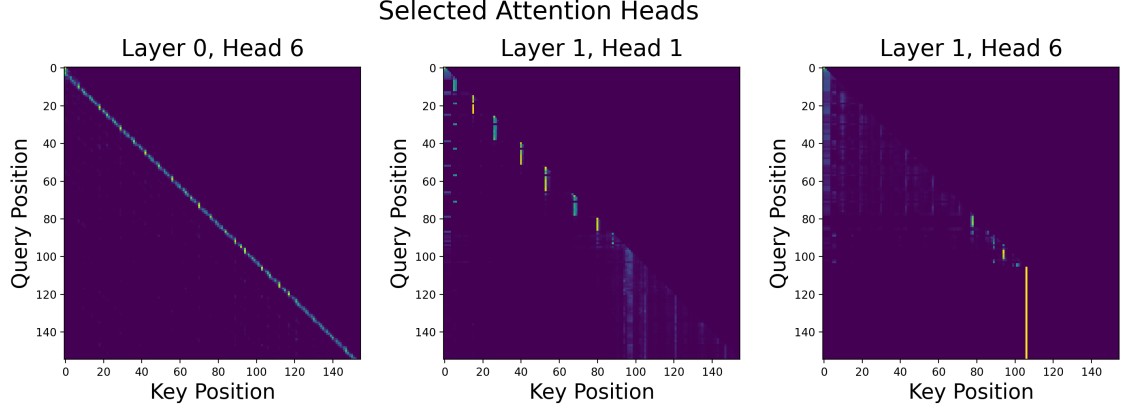

Figure 10: Attention score visualization of selected attention heads from our 6-layer transformer model with embedding dimension 768. The visualization shows attention patterns for a tokenized polynomial sequence and its decomposition.

**Input polynomial:** $+ * P\ 2\ 5\ 6 \wedge a0\ P\ 9 + * N\ 1\ 9\ 2 * \wedge a0\ P\ 8\ a1 + * P\ 4\ 8 * \wedge a0\ P\ 7 \wedge a1\ P\ 2 + * N\ 4 * \wedge a0\ P\ 6 \wedge a1\ P\ 3 + * N\ 6\ 4 * \wedge a0\ P\ 3 \wedge a1\ P\ 6 + * P\ 1\ 6 * \wedge a0\ P\ 2 \wedge a1\ P\ 7 * P\ 6\ 4 \wedge a1\ P\ 9\ ?$

**Model's decomposition output:** $+ * N\ 4 \wedge b0\ P\ 3 + * b0 \wedge b2\ P\ 2 * N\ 1 \wedge b2\ P\ 3\ \& + * N\ 4 \wedge a0\ P\ 3 * \wedge a0\ P\ 2\ a1\ \& + * N\ 3 \wedge a1\ P\ 3 + * N\ 2 * a1 \wedge a2\ P\ 2 * N\ 4 \wedge a2\ P\ 3\ \& * N\ 4 \wedge a1\ P\ 3$ The visualization reveals how different attention heads focus on specific structural elements when decomposing polynomials.