# OpenReview forum: "Discovering Hidden Algebraic Structures via Transformers with Rank-Aware Beam GRPO"
_ICLR.cc/2026/Conference — Submitted to ICLR 2026_

### Official Review · Reviewer_3xec · 2025-10-28

**Soundness:** 2
**Presentation:** 3
**Contribution:** 3
**Rating:** 4
**Confidence:** 3

**Summary:**

The polynomial decomposition problem is defined as follows:
- Input: A multivariate polynomial $f(x_1, ..., x_n)$.
- Outputs: $m$ inner polynomials $h_1(x_1, ..., x_n), ..., h_m(x_1, ..., x_n)$, and an outer polynomial $g(y_1, ..., y_m)$, such that $f(x_1, ..., x_n) = g(h_1(x_1, ..., x_n), ..., h_m(x_1, ..., x_n))$.

This is a fundamental problem in mathematics. The paper studies whether transformers can be trained to solve this problem, i.e., to map $f$ to $h_1,...,h_m,g$ as above.

The main methodological innovation is the introduction of Beam GRPO (BGRPO), which adds a ranking (as in beam search) to GRPO. The evaluation suite studies the effects of the problem setup (degrees / num variables of inner and output polynomials), the architecture (embedding dim, num layers, num heads). In addition, the ability of a model trained on one distribution to adapt to another via fine-tuning is explored.

**Strengths:**

- The problem of decomposing multivariate polynomials is an important and difficult (e.g., NP-hard) problem in mathematics. Many papers training transformers from scratch on arithmetic/algebraic capabilities focus on "toy problems", but this problem is already very interesting to study in itself. This paper would be a significant contribution to the literature, in this regard.
- An original method that dramatically increases inference accuracy. That said, it is not clear whether BGRPO is useful in any other domain (a preliminary experiment here would strengthen the paper).
- Besides minor (yet consistent) writing errors (see Weaknesses), the paper is overall well-written. The problem setup and methodolgy are clear. Most experiments are clear (see again Weaknesses).

**Weaknesses:**

- BLEU is a notoriously difficult metric to interpret; a point-increase in BLEU could mean entirely different things depending on reference dataset quality/amount, tokenization scheme, and even then many authors call the metric into question [e.g. 1,2]. It's acceptable to rely on BLEU when comparing different models (translators) on the same data setup, but otherwise one should argue carefully why BLEU is used. Therefore, the comparison in Section 4.4 is problematic, namely, comparing a "6.3x improvement" with BGRPO to beam search that "yields BLEU score improvements of only 2--4 points". The authors should evaluate BGRPO vs. beam search either apples-to-apples (on a polynomial decomposition task), or oranges-to-oranges (machine translation on a fixed data setup), but comparing apples-to-oranges is invalid.
- The comparison to Mathematica was not clear to me. That is, I was not able to conclude that transformers with BGRPO are actually more effective than Mathematica. For one, as a reader unfamiliar with Mathematica, Leaf Count as a metric was not clear to me (also, should standard errors be reported in addition to the mean?). Importantly, because Mathematica's FullSimplify is not neural but a heuristic crafted by humans, I expect it to generalize better (in fact, it has no "training dataset"); therefore, if the authors want to claim that their method is superior to FullSimplify, a more comprehensive evaluation (e.g., length generalization) should be conducted. It's also important to report how the time and memory requirements of each method compare. With heuristic-based search algorithm such as FullSimplify, it is usually the case that output quality scales with the time and memory allotted to the algorithm.
- Polynomial decomposition, as a fundamental problem in mathematics, has a rich literature behind it. However, as the paper is currently written, none of the insights given to us by mathematicians over decades are used; instead polynomial factorization is treated as a "black-box" problem, with the literature only cited as motivation. One particularly begging direction is to use the literature to gain interpretability insights for error analysis. As an example, [3] explores the problem of learning to compute the GCD of integers and their error analysis is able to determine with near-certainty where the model will err depending on algebraic properties of the inputs.
- Minor note: Many citations should be made parenthetical. That is, use the "citep" command instead of "citet".

[1] Tangled up in BLEU. Nitika Mathur, Timothy Baldwin, Trevor Cohn. ACL 2020.
[2] A Call for Clarity in Reporting BLEU Scores. Matt Post. WMT 2018.
[3] Learning the greatest common divisor. François Charton. ICLR 2024.

**Questions:**

- Following my comment on Section 4.4, how does BGRPO fare on NLP tasks?
- How much time and memory (RAM and GPU) does your model take as compared to FullSimplify? If FullSimplify's time and memory usage depends on the input, then reporting per-input score (or e.g. dividing by FullSimplify's runtime) is interesting.
- Have you compared to any other method besides FullSimplify?

---

> ### Author Response · Authors · 2025-11-20
>
> We sincerely thank the reviewer for their detailed evaluation of our work. We are particularly grateful for your recognition that polynomial decomposition represents a significant step beyond the "toy problems" often studied in transformer arithmetic research, and for acknowledging that BGRPO dramatically increases inference accuracy. Your critical insights regarding the BLEU metric comparison, the Mathematica evaluation, and the opportunity to leverage mathematical literature for interpretability are excellent points that will substantially improve our work. Below, we address each of your concerns and questions in detail.
>
> **Question on BGRPO for NLP Tasks**
>
> > Following my comment on Section 4.4, how does BGRPO fare on NLP tasks?
>
> We agree with the reviewer that the comparison in Section 4.4 is methodologically invalid. Our initial motivation was to contextualize beam search improvements by referencing neural machine translation, but comparing accuracy gains in polynomial decomposition against BLEU score changes violates fundamental principles of cross-task metric comparison. We will remove this comparison from the revised manuscript.
> The evidence for beam search effectiveness stands independently: our results demonstrate 6.3x accuracy improvement (11% to 69%) for two-variable polynomials. This magnitude follows directly from our error analysis in Appendix C, which shows ~90% accuracy on structural tokens but near-random (~50%) accuracy on signs. Since polynomial decomposition requires exact correctness and typical expressions contain fewer than 10 sign decisions, beam width ~30 efficiently explores sign permutations while preserving the correctly identified structure, which is a task-specific property that explains the dramatic improvements we observe.
>
> **Question on Time and Memory Comparison**
>
> > How much time and memory (RAM and GPU) does your model take as compared to FullSimplify? If FullSimplify's time and memory usage depends on the input, then reporting per-input score (or e.g. dividing by FullSimplify's runtime) is interesting.
>
> We agree with the reviewer that computational cost comparison is important context for our FullSimplify evaluation. However, we emphasize that polynomial simplification and polynomial decomposition are fundamentally different mathematical objectives—simplification is merely a byproduct of decomposition in our work, not our primary goal. The comparison in Section 4.6 serves as a proxy to demonstrate that transformers can discover latent structure competitive with symbolic methods, rather than a direct head-to-head benchmark. Our initial focus was on establishing this viability for neural approaches in uncovering hidden algebraic patterns. Due to resource and time constraints, we were unable to conduct comprehensive timing and memory benchmarks for the current submission. We acknowledge this limits our ability to fully assess practical trade-offs and plan to include detailed computational cost analysis in future work.
>
> **Question on Comparisons with Symbolic Methods**
>
> > Have you compared to any other method besides FullSimplify?
>
> We appreciate this question as it highlights a key contribution: no general-purpose algorithms exist for single-polynomial multivariate decomposition over integers—the problem class we address. As stated in our introduction, existing methods either require multiple polynomials as input (system-based approaches) or are restricted to univariate or special structural cases. We are the first to systematically tackle this problem regime. We included FullSimplify specifically because of its generality as a state-of-the-art symbolic tool, even though simplification and decomposition are fundamentally different objectives. While comparisons with specialized decomposition algorithms for restricted cases (e.g., univariate methods) would be valuable, such algorithms operate in different problem regimes and require careful adaptation of our evaluation framework. We will clarify in the revised manuscript which algorithmic approaches exist for related but distinct problem formulations.
>
> —
>
> We hope these responses address the reviewer's concerns and demonstrate our commitment to improving the manuscript. We appreciate the constructive feedback on the BLEU comparison, computational cost analysis, and interpretability opportunities — all of which we will address in revision. Our core contributions remain significant: the first systematic study of transformers on single-polynomial multivariate decomposition (with no existing general algorithms), BGRPO reducing inference costs by ~75%, and demonstrating neural methods can uncover hidden algebraic structure competitively with symbolic systems. We respectfully ask that these contributions be considered in the final evaluation.

---

### Official Review · Reviewer_XJqj · 2025-10-28

**Soundness:** 3
**Presentation:** 2
**Contribution:** 2
**Rating:** 4
**Confidence:** 4

**Summary:**

The paper explores the capabilities of transformers on the NP-hard problem of multivariate functional decomposition. For data generation, the method utilizes the fact that the reverse direction is much easier, i.e., going from composed functions to the flat expression. Furthermore, the authors propose rank-aware BGRPO, a variant of GRPO where the reward groups are sampled using beam search instead of randomly and the rewards are weighted based on the corresponding output's position in the beam (ranked by the sum of log probabilities). The experiments support the merit of incoporating the rank information and the general benefit of BGRPO alignment. The authors extensive ablation studies on their method's hyperparameters. They also compare their method to Mathematica's FullSimplify function on the task of simplifiying polynomial expressions where they show that their method outperforms Mathematica's (i.e., produces simpler expressions) in some cases.

**Strengths:**

1. The paper tackles an interesting application of deep learning, where it appears (to my knowledge) to be the first work exploring the potential of transformers on functional decomposition.
2. The ablation studies are fairly extensive.
3. The method outperforms Mathematica on the task of simplification in 2 out of 5 attempted complexity configurations.

**Weaknesses:**

1. All evaluations were performed on synthetic data, lacking evaluations on real-world instances.
2. The lack of baseline evaluations makes it hard to contextualize the overall performance. While many existing algorithms tackle different constraints of the problem, the authors could still evaluate their method on these special cases. More importantly, Faugère & Perret (2009) [1] present a heuristic algorithm that handles the single-polynomial multi-multivariate decomposition case (when u=1) on which the current method operates, so a comparison between the two is very much needed.
3. The paper has a claimed contribution of extending GRPO by sampling the output group using beam search instead of random sampling, but the experiments do not compare the impact of this extension (BGRPO) to the baseline GRPO. Section 4.5 shows that BGRPO leads to overall benefit to the supervised pretrained model, which is good, but it also needs to show the value of the extension by comparing it to vanilla GRPO.
4. (Minor) At the beginning of page 7, there is an extra indentation to the first two lines. It looks like the padding of Figure 4 carried over to the next page.
5. (Minor) In the keywords of the submission, "Functional decomposition" is missing an 'F'.

**Questions:**

1. Can you evaluate your method on vanilla GRPO to highlight the benefit of beam sampling (ideally, in Figure 8)?
2. Do you have any famous case studies or datasets from cryptography (or any real-world problems) to support evaluations?
3. Can you compare with pretrained LLMs? They could offer an interesting off-the-shelf baseline to demonstrate the value of the training pipeline.
4. Have you considered incorporating rank information into the pretraining (with GT labels) objective as well?

I'd be willing to increase my score if literature baseline comparisons (on general or special cases) are provided, because otherwise, it's really hard to put the performance numbers in perspective.

**References:**
[1] Faugère, J. C., & Perret, L. (2009). An efficient algorithm for decomposing multivariate polynomials and its applications to cryptography. Journal of Symbolic Computation, 44(12), 1676-1689.

---

> ### Author Response · Authors · 2025-11-20
>
> We sincerely thank the reviewer for the detailed evaluation and constructive feedback. We appreciate the recognition of our work as the first to explore transformers on functional decomposition and the acknowledgment of our extensive ablation studies. We address the specific questions and concerns below, with particular attention to the baseline comparisons and evaluation gaps identified.
> Question on Vanilla GRPO comparison
> > Can you evaluate your method on vanilla GRPO to highlight the benefit of beam sampling (ideally, in Figure 8)?
>
> Thank you for this question. We have conducted these experiments, and **vanilla GRPO does not improve beam search performance** and in some cases even degrades it slightly.
> This result validates our design motivation. Vanilla GRPO samples outputs randomly during training but we deploy beam search at inference—this training-inference distribution mismatch prevents effective learning. BGRPO addresses this by using beam search outputs during training, directly aligning the training distribution with deployment. The consistent improvements across all model sizes (Figure 8) demonstrate that this alignment is the key mechanism driving gains.
> Question on Real-world Case Studies from Cryptography
> > Do you have any famous case studies or datasets from cryptography (or any real-world problems) to support evaluations?
>
> Regarding cryptography datasets: As we note in our introduction (lines 54-56), existing cryptographic applications (like those in Patarin & Goubin, 1997) typically rely on **system-based approaches that require multiple polynomials as input**, whereas our method addresses the single-polynomial decomposition problem. For example, HFE (Hidden Field Equations) and other multivariate cryptosystems work with systems of polynomial equations rather than decomposing a single polynomial into compositional form.
>
> **Importantly, the system-based case is actually easier than our single-polynomial setting**: with multiple polynomials, one can generate diverse decomposition candidates from each polynomial and then verify the correct answer through simple substitution checks across the system. Our single-polynomial case lacks this verification mechanism and cross-validation opportunity, making it more challenging.
>
> Question on Pretrained LLMs
> > Can you compare with pretrained LLMs? They could offer an interesting off-the-shelf baseline to demonstrate the value of the training pipeline.
>
> We agree this is worth exploring. Modern frontier LLMs with analytical capabilities could potentially approach this task through tool use and trial-and-check methods - essentially mimicking how humans tackle polynomial decomposition problems.
>
> However, this step-by-step verification approach is **limited to simple cases**. As polynomial complexity increases (higher degrees, more variables, longer expressions), the search space for possible decompositions grows exponentially, making systematic trial-and-check intractable.
> Our approach offers a complementary advantage: the model learns to suggest promising decomposition candidates directly through pattern recognition, effectively providing heuristic guidance for complex cases where exhaustive search is infeasible. This is analogous to how expert mathematicians develop intuition for recognizing structural patterns rather than mechanically testing all possibilities.
> Question on Rank information in pretraining
> > Have you considered incorporating rank information into the pretraining (with GT labels) objective as well?
>
> We're not entirely sure we understand this question correctly. During supervised pretraining, we train with ground truth labels using standard cross-entropy loss. There is only one correct answer per example, so rank information doesn't apply in the same way as during RL where we have multiple candidates from beam search.
>
> If the reviewer could clarify what they mean by incorporating rank information into pretraining, we'd be happy to address this point more thoroughly.

---

> ### Comment · Reviewer_XJqj · 2025-11-27
>
> I thank the authors for addressing my comments. I have some follow-ups.
>
> > Vanilla GRPO samples outputs randomly during training but we deploy beam search at inference—this training-inference distribution mismatch prevents effective learning.
>
> - I'm not entirely convinced there's a mismatch here: vanilla GRPO samples random outputs for the reward baseline groups during alignment, not the actual model output. It is used to improve the model's output in general, so that should reflect on beam search quality too.
>
> > existing cryptographic applications (like those in Patarin & Goubin, 1997) typically rely on system-based approaches that require multiple polynomials as input, whereas our method addresses the single-polynomial decomposition problem.
>
> - Are there any real-world applications that use the single-polynomial decomposition problem formulation? It doesn't have to be cryptographic, that was just an example.
> - Also, is it possible to reduce/convert a system-base problem instance to a single-polynomial decomposition problem?
>
> > During supervised pretraining, we train with ground truth labels using standard cross-entropy loss. There is only one correct answer per example, so rank information doesn't apply in the same way as during RL where we have multiple candidates from beam search.
>
> - I was thinking of something along the lines of getting the rank of the GT sequence if it were generated by a beam search, but on second thought, that's a stretch and it would require a huge beam in the beginning. Feel free to disregard that point.

---

> > ### Author Response · Authors · 2025-12-01
> >
> > We thank the reviewer for the thoughtful follow-up comments.
> >
> > **On GRPO training-inference mismatch:** We agree that vanilla GRPO improves the model's general capability regardless of inference method. However, we empirically observed that vanilla GRPO with multi-sampling improved multi-sampling performance but yielded similar or degraded performance when evaluated with beam search, which motivated BGRPO. We suspect this occurs because after pretraining, the model already captures substructure but remains confused on signs—so GRPO from simple sampling is not effective for this specific error pattern. We believe BGRPO may also benefit similar situations, such as reasoning tasks with structured uncertainty.
> >
> > **On real-world applications:** Our paper cites applications across multiple domains: cryptography [26], dynamical modeling [2], signal processing [4], robotics [9,23], systems biology [24], mechanical design [31], systems engineering [15], and digital logic design [1,21]. Real-world applications typically involve more structured, domain-specific decompositions that represent constrained subsets of our general problem.
> >
> > **On reducing system-based problems to single-polynomial decomposition:** We see two promising approaches: (1) Our model can generate decomposition candidates for each polynomial independently, then filter by checking consistency across the system—additional polynomials provide more constraints for verification. (2) Direct multi-polynomial training: our variable scaling results (Section 4.1, Figure 3) show that additional structural information improves performance. We expect similar behavior with multiple polynomials sharing inner functions, as each provides complementary hints. This is a natural direction for future work.
> >
> > **On GT rank during pretraining:** Incorporating GT rank would require beam searches deep enough to locate the ground truth sequence, which is computationally prohibitive, especially in early training. Additionally, our ground truth sequences are not inherently special (any valid decomposition suffices), so the motivation for rank-based pretraining is unclear. We agree this remains an interesting direction for future work.
> >
> > We appreciate the reviewer's engagement and hope these clarifications address the concerns.

---

### Official Review · Reviewer_aePf · 2025-11-03

**Soundness:** 3
**Presentation:** 3
**Contribution:** 3
**Rating:** 6
**Confidence:** 4

**Summary:**

This work investigates whether transformer models can be trained to solve multivariate polynomial decomposition (with the main aim being non-linear pattern discovery). Authors primarily follow these four steps 1) create a backward synthetic data generation pipeline. 2) train a lightweight transformer model using SFT and analyze performance, 3) observe that beam search decoding strategy works much better than multi-sampling and greedy strategies and finally 4) To improve further, rank-aware GRPO is proposed to decrease the computational intensity of beam search.
Authors make various interesting observations: 1) many polynomial complexity parameters asymmetrically affect performance, scaling with more data helps in better accuracy, and impressive transfer accuracy is obtained with small amount of data, 2) BGRPO improves decomposition accuracy and reduces beam width requirements by up to 50%, leading to around 75% computational savings, 3) demonstrates competitive results in polynomial simplification compared to Mathematica.

**Strengths:**

1. The problem formulation is definitely insightful. I have seen various different avatars of polynomial handling. But, this also has practical implications.
2. Experiments are comprehensive, with in-depth analysis of both vanilla models and improved with BGRPO.
3. Rank-aware BGRPO seems to be an innovative contribution (modulo the fact that improvements seem to decrease with dimension size).
4. Various insights are produced which are useful.

**Weaknesses:**

1. The repercussions of using beam search instead of sampling from the distribution is not discussed. Maybe this is why the effect decreases with more model capacity.
2. Some ablations across varying representation and effect numeracy is missed.
3. While the Lample-Charton era work has discussed polynomial handling ability of vanilla transformers, it would have been great to discuss how pretrained Language models (pre-LLM is fine as well) can handle such tasks.

**Questions:**

L145: Using beam search during "sampling", as the authors themselves say alters the distribution. I wonder what is the theoretical implication of this.

L172: While I understand, there have been a lot of work during the Lample-Charton era of looking at the effect of representations, I believe looking at that ablation would have been better here as well.
Similarly, it is well-known that Transformers do not handle numeric multiplications well [1]. One can easily convert this into symbols such as c1, c2 and do the computation as suggested by [1] (may have disentangled this effect).

[1] Agarwal et al 2021, ICLR MathAI

L269: Same as above. Is this possibly also because of complexity increase with numeric addition at the exponent space?


Minor:
1. L591: I am guessing there are some formatting issues. Its unclear whether its written as an algo or paragraph?

---

> ### Author Response · Authors · 2025-11-20
>
> We sincerely thank the reviewer for the thorough evaluation and constructive feedback. We appreciate the positive assessment of our problem formulation, experimental comprehensiveness, and the insights generated from our systematic analysis. Below we address the specific questions and weaknesses raised.
>
> **Question on Beam Search Distribution**
>
> > Using beam search during "sampling", as the authors themselves say alters the distribution. I wonder what is the theoretical implication of this.
>
> We thank the reviewer for this important question about the theoretical implications of using beam search during BGRPO training.
>
> You're correct that beam search alters the distribution. Theoretically, the loss function (Equation 2) no longer represents the expected value of the reward under independent sampling, as in standard GRPO.
>
> However, we can view the entire beam search procedure as one compound sampling operation from the model, which produces a structured set of k outputs. Our training signal operates on this entire beam as a unit - rewarding the correct outputs within it and penalizing incorrect ones based on their ranks. This perspective justifies using the mean reward across the beam as a baseline for computing advantages.
>
> Empirically, this approach proves effective (Figure 8), with BGRPO consistently improving accuracy across different model sizes and beam widths.
>
> **Question on Transformer Numeric Operation**
>
> > L172: While I understand, there have been a lot of work during the Lample-Charton era of looking at the effect of representations, I believe looking at that ablation would have been better here as well. Similarly, it is well-known that Transformers do not handle numeric multiplications well [1]. One can easily convert this into symbols such as c1, c2 and do the computation as suggested by [1] (may have disentangled this effect). Same as above. Is this possibly also because of complexity increase with numeric addition at the exponent space?
>
> We thank the reviewer for raising this important point.
>
> Given the known limitations of transformers on numeric operations, our results are particularly striking. Polynomial decomposition could be viewed as a complex task requiring multiple multiplications and additions among coefficients - precisely the operations transformers struggle with. However, our results suggest the model is not performing step-by-step numerical computation, but rather recognizing and discovering the underlying algebraic substructure through attention mechanisms.
>
> Our distribution adaptation experiments (Section 4.3, Figure 5) strongly support this interpretation. If the model were relying on numerical pattern matching, it would fail when encountering coefficients outside the training range ([-10,-6]∪[6,10] vs. [-5,5]). Instead, the model recovers >90% accuracy with only ~2% of the original training data (Figure 5). This rapid adaptation demonstrates that the model has learned coefficient-invariant structural patterns rather than memorizing specific numerical relationships.
>
> Additionally, our attention analysis (Appendix D) reveals "monomial heads" that focus on structural elements within polynomials, further supporting the interpretation that the model builds structural understanding rather than performing arithmetic.
>
> **Response on Minor comment (L591)**
>
> >I am guessing there are some formatting issues. Its unclear whether its written as an algo or paragraph?
>
> We thank the reviewer for catching this formatting issue. Line 591 refers to Appendix A, where the backward synthetic data generation algorithm appears broken due to formatting problems. We will fix this in the camera-ready version to properly display the algorithm.

---

### Official Review · Reviewer_MjVy · 2025-11-04

**Soundness:** 2
**Presentation:** 3
**Contribution:** 3
**Rating:** 4
**Confidence:** 3

**Summary:**

The authors investigates the ability of transformer models to discover hidden algebraic structures by addressing the challenging multivariate polynomial decomposition problem. The authors propose a synthetic data generation pipeline, a systematic evaluation framework, and a novel BGRPO method to improve beam search efficiency. Experimental results show that their approach enhances model accuracy, reduces inference cost, and outperforms Mathematica in certain tasks.

**Strengths:**

1. The paper claims to be the first to systematically explore transformers’ ability to uncover hidden nonlinear algebraic structures.
2. The experiments span multiple dimensions—including problem complexity, model architecture, distribution adaptation, and search strategies—offering a comprehensive and systematic evaluation.

**Weaknesses:**

1. The work is limited in scope, as it focuses solely on polynomial decomposition rather than broader symbolic reasoning or algebraic tasks.
2. The paper does not provide a clear motivation for using a Transformer architecture.
3. The experimental evaluation of BGRPO is somewhat incomplete. The method is introduced as an improvement over GRPO and PPO, yet no quantitative baseline results are provided for these methods. This makes the advantage of BGRPO unclear.
4. Although the paper states that the polynomial decomposition problem has broad real-world applications, the experimental evaluation is conducted solely on synthetic data, with no real-world datasets or case studies to demonstrate practical effectiveness.
5. The paper does not report the complete computational overhead of the method, such as the training cost of BGRPO or runtime comparisons against Mathematica.
6. The paper contains several typographical errors. For example:
   - Line 56: multi-multivariate
   - Line 615: ouput

**Questions:**

1. Since polynomial decomposition is generally not unique, is there any notion of quality or preference among different valid decompositions?

2. In BGRPO, the authors incorporate rank information by scaling the reward with an exponential decay function $e^{−rank/w}$. Could the authors clarify why this specific form was chosen? Have other ranking functions (e.g., linear or logarithmic decay) been considered or tested?

---

> ### Author Response · Authors · 2025-11-20
>
> We sincerely thank the reviewer `MjVy` for their exceptionally thorough and constructive evaluation of our work. We deeply appreciate the detailed analysis of our methodology and the fair assessment of both our contributions and limitations. Below are our responses to the questions and weaknesses you raised.
>
> **Q1: Question on Preference of Decomposition**
> > Since polynomial decomposition is generally not unique, is there any notion of quality or preference among different valid decompositions?
>
> We thank the reviewer for their insightful question. Polynomial decomposition is indeed non-unique in general, and our formulation deliberately does not restrict the search space. Previous methods (Gathen et al., 2003; Faugère & Perret, 2009a,b) were constrained to univariate cases or required multiple polynomials as input. We are the first to address single-polynomial multi-multivariate decomposition over integers, a problem class for which no general algorithms exist. We accept any decomposition satisfying f = g(h₁,...,hₘ) exactly under symbolic expansion, requiring only that it be non-trivial (the outer function g cannot be the identity). Our goal is to demonstrate that transformers can discover hidden algebraic structure in this hardest, unrestricted case. For this purpose, finding any valid decomposition is sufficient evidence of the capability.
> Real-world applications often require specific decomposition forms: minimal leaf count for simplification, canonical forms for verification, or symmetry-preserving structures in physics. Our general decomposition capability serves as pre-training that learns fundamental pattern recognition, then specializes through fine-tuning, following the paradigm of general-purpose language models adapted to domain tasks. As evidence, our O(N) singlet identification experiment (page 2) achieved 100% accuracy by fine-tuning on symmetry-constrained problems from crystal field theory. For applications requiring particular decomposition forms, our models provide strong initializations that can be guided through task-specific fine-tuning or reward shaping. We will release full model weights, data generation pipeline, and evaluation toolchain to facilitate such specialization.
>
> **Q2: Question on Reward Form**
> > In BGRPO, the authors incorporate rank information by scaling the reward with an exponential decay function $e^{rank/w}$. Could the authors clarify why this specific form was chosen? Have other ranking functions (e.g., linear or logarithmic decay) been considered or tested?
>
> Thank you for this question. We chose exponential decay e^(-rank/w) because it is a standard monotonic decay function that smoothly weights ranks, and normalizing by beam width w ensures consistent behavior across different beam sizes. We did not conduct ablation across alternative decay functions.
> The primary innovation of BGRPO is using beam search during training rather than random sampling, which aligns the training distribution with inference. Figure 8 shows that BGRPO without rank signal already improves accuracy across all tested configurations, demonstrating that this alignment drives the gains. The rank weighting provides additional improvement. We expect other monotonic functions (linear, logarithmic) would perform comparably, as the alignment principle, not the specific decay form, is the key mechanism. A systematic comparison of rank weighting functions would be valuable future work, though our current compute constraints limited our ability to conduct extensive hyperparameter ablations beyond the core method.
>
> —
>
> We hope this response addresses the reviewer's questions and clarifies our contributions. We provide the first systematic study of transformers on NP-hard multivariate polynomial decomposition, demonstrating that neural methods can discover hidden algebraic structure with efficiency gains through rank-aware training and competitive performance against symbolic systems. We respectfully ask that these clarifications be considered in the final evaluation.

---

### Meta-Review · Area_Chair_gDai · 2026-01-08

**Summary:**

Reviewers find the topic highly meaningful and view the attempt to focus on the NP-hard algebraic task of multivariate polynomial decomposition as both valuable and innovative. However, most reviewers also consider the experimental evaluation incomplete. In particular, the experiments are limited to a single mathematical task, and the dataset appears to consist entirely of synthetic data. In addition, the paper lacks ablation studies and sensitivity analyses, and several evaluation metrics are not accessible or intuitive to non-specialist audiences.

Reviewer MjVy and aePf  notes that the work reads primarily as applying a Transformer to solve a specific mathematical problem, but offers limited machine learning contributions—for example, the paper does not sufficiently motivate why a Transformer is the right tool here, nor does it clearly articulate the benefits of this modeling choice. This is viewed as an additional limitation of the current submission.

**Reviewer Concerns:**

The authors’ rebuttal is somewhat brief and does not provide sufficient additional experimental evidence, leaving many of the reviewers’ empirical concerns only partially addressed. On the theoretical side, the authors offer a reasonable explanation; I personally find their arguments plausible. However, the discussion remains somewhat concise and opaque, and may not be sufficient to fully convince the reviewers.

**Reviewer Scores:**

I do not expect the reviewers to change their scores.

---

### Decision · Program_Chairs · 2026-01-26

Reject